# Towards a great ape dictionary: Inexperienced humans understand common nonhuman ape gestures

**Kirsty E. Graham** [ID]*, Catherine Hobaiter*

Wild Minds Lab, School of Psychology and Neuroscience, University of St Andrews, St Andrews, Scotland, United Kingdom

* keg4@st-andrews.ac.uk (KEG); clh42@st-andrews.ac.uk (CH)

## Abstract

In the comparative study of human and nonhuman communication, ape gesturing provided the first demonstrations of flexible, intentional communication outside human language. Rich repertoires of these gestures have been described in all ape species, bar one: us. Given that the majority of great ape gestural signals are shared, and their form appears biologically inherited, this creates a conundrum: Where did the ape gestures go in human communication? Here, we test human recognition and understanding of 10 of the most frequently used ape gestures. We crowdsourced data from 5,656 participants through an online game, which required them to select the meaning of chimpanzee and bonobo gestures in 20 videos. We show that humans may retain an understanding of ape gestural communication (either directly inherited or part of more general cognition), across gesture types and gesture meanings, with information on communicative context providing only a marginal improvement in success. By assessing comprehension, rather than production, we accessed part of the great ape gestural repertoire for the first time in adult humans. Cognitive access to an ancestral system of gesture appears to have been retained after our divergence from other apes, drawing deep evolutionary continuity between their communication and our own.

## Introduction

Regarded by philosophers and scientists alike as the cognitive capacity most critical to human uniqueness [1], the apparent discontinuity between human language and nonhuman communication has been argued to present an evolutionary puzzle. However, more and more research has started to unveil language's deep phylogenetic roots: from the way other species combine signals to change the meaning (we use "meaning" in this article to refer to signal functions and Apparently Satisfactory Outcomes; [2,3]) of an utterance [4]; to their use of social inference in communication [5]; to how behavioural and social contexts seem to disambiguate signal meanings [6]. Nevertheless, many species' communication is based on the exchange of specific, detailed information: Alarm calls, for example, can encode combinatory information on both the type and proximity of a predator [7,8]. While a rich source of information, these signals

**Data Availability Statement:** Data and code are available in an open access repository at: https://doi.org/10.5281/zenodo.7347608. The experiment was presented in Gorilla.sc (www.gorilla.sc); a full preview and all importable sheets are available

through Gorilla Open Materials (https://app.gorilla.sc/openmaterials/344409); and video data files are available at the Great Ape Dictionary on Youtube (https://tinyurl.com/greatapedictionary).

**Funding:** This research received funding from the European Union's 8th Framework 287 Programme, Horizon 2020, under grant agreement no 802719 to CH (https://ec.europa.eu/info/research-and-innovation/funding/funding-opportunities/funding-programmes-and-open-calls/horizon-2020_en). This work was supported by Gorilla Awards in Behavioural Science who provided the Gorilla.sc licensing fee and an unlimited participant award to KG (https://gorilla.sc/). The funders had no role in study design, data collection and analysis, decision to publish, or preparation of the manuscript.

**Competing interests:** The authors have declared that no competing interests exist.

typically exist as a fixed response to stimuli, produced irrespective of a recipient's attention or interest, or even whether a recipient is there [9]. Humans produce these types of signals too. Picking up a too-hot pan from the cooker, we might give an involuntary yelp, shake our hand, and/or make a facial grimace of pain. Any potential recipients around receive useful information from these signals: The pan is hot! But we did not yelp, shake, or grimace with the goal of communicating, we'd have done it whether someone was there or not. Language is different. We choose whether to tell someone who was out of the room to "watch out, the pan is hot." We can use it in the absence of the stimuli that we were originally responding to. We would stop using it once our recipient indicated that they understood. We might even use it to talk to ourselves.

Fundamentally, with language, we do more than broadcast information; we intend to communicate a goal to a partner we recognise as having their own behaviour, goals, and knowledge. Human languages' *intentional* nature takes it beyond sharing information: It communicates meaning [10–13]. This fundamental property is very rarely observed in other species [9,14], and when it is, it is typically restricted to one or two signals used in a highly specified way [15,16]. Nevertheless, the emergence of intentional communication through a single recent genetic leap in the human lineage remains implausible; instead, precursor abilities were likely present in the communication of our evolutionary ancestors and should be shared among modern ape species today [17].

Strikingly, great ape gestures are used in this language-like way: Rich *systems* of over 80 signals deployed communicate everyday goals (for example, [2,18–25]), and ape gesturing has been suggested to be an important scaffold in the evolution of human language [26,27]. Great ape repertoires show substantial overlap across species, including overlap among ape species more distantly related than chimpanzees, bonobos, and humans [28–32]. As a result, we would expect humans to retain the use of this system of ape gestural communication; but, to date, the use of naturalistic ape gestures appeared to be absent in human communication. Humans are highly gestural, deploying deictic, iconic, conventional, co-speech/co-sign, among other kinds of gestures. However, this itself is part of what makes studying gestural overlap between adult humans and other apes challenging. Gestures shared with other apes may be masked by the myriad ways that people signal with their hands and body. From pointing to pantomime, language-competent humans regularly employ gestures that accompany [33] and may even create [34] language; highly variable across cultures, they are rarely used to independently convey the core goal of the communication and do not map closely onto those employed by nonhuman apes. Unpicking gestures from the great ape repertoire in naturalistic adult human gesturing may not be impossible, but it will take a substantial collaborative effort to one day do so. In the meantime, there are other methods at our disposal. A recent study suggested that gestures from the "ape repertoire" may not be completely absent: Before language emerges, preverbal 1- to 2-year-old human infants were found to deploy over 50 gestures from the ape repertoire [35]. Given the available movements and body parts, there are well over 1,000 potential gesture forms that could be produced with the ape body, but apes only use approximately 12% of these [36]. Thus, any overlap between species is very unlikely to be trivial. Here, we provide the first test of the hypothesis that language-competent adult humans still share access to "family-typical" great ape gesture.

We employ a method regularly used in studies of nonhuman primate communication, a "play-back" experiment, in which recipient behaviour is analysed following exposure to a signal [37,38]. This type of comprehension study has historically been employed to test nonhuman species on comprehension of human language [39,40], but here we flip the paradigm to test humans on nonhuman communication. Of course, our experimental paradigm is more conventional in the human psychology literature and has the advantages that, with humans rather than nonhumans, we are able to conduct tests with untrained participants and to use

text responses in this match-to-sample type paradigm. While language-competent humans seem to no longer typically produce gestures from the ape repertoire (or that these gestures may be masked by other common human-typical gesturing), the presence of a signal in an individual's communicative repertoire can also be shown through their comprehension of it [30]. We conducted an online experiment to crowdsource whether adult human subjects understand the meaning of gestures produced by nonhuman apes. The experiment was presented in Gorilla.sc (www.gorilla.sc; [41]); a full preview and all importable sheets are available through Gorilla Open Materials (https://app.gorilla.sc/openmaterials/344409); and video data files are available at the Great Ape Dictionary on YouTube (https://tinyurl.com/greatapedictionary). Participants were randomly allocated to two conditions: those who viewed gesture videos only (Video only), and those who viewed gesture videos with a brief, one-line description of context (Context). Each video was accompanied by a simple illustration of the gesture to assist inexperienced viewers in identifying the gesture action (https://greatapedictionary.ac.uk/gesture-videos2/). From a set of 40 videos, each participant saw 20 videos with examples of ape gesture (10 chimpanzee, 10 bonobo). Videos were cut to show only the gesture, eliminating any behaviour before or after communication.

We selected the 10 most common gesture types for which we were previously able to confirm "meaning" in both chimpanzees and bonobos, determined by recipient responses that consistently satisfy the signaller [19]. Chimpanzees and bonobos are humans' closest living relatives (we are also theirs, with the split from humans more recent than the last common ancestor shared between *Pan* and *Gorilla*; [42]). While in principle, given the overlap in gesture repertoires across all apes [28], we would predict that gorilla and orangutan gestures may also be salient to humans, the meanings for gestures in these ape species are not yet established.

Some gestures are used towards a single meaning (i.e., recipients consistently respond in the same way to that gesture), whereas others are used towards two or more meanings [2,19]. For example, the Big Loud Scratch is used to initiate grooming (meaning = "Groom me"), while Object Shake is used to initiate copulation (meaning = "Let's have sex"), to initiate grooming (meaning = "Groom me"), and to increase distance between signaller and recipient (meaning = "Move away"). The correct meaning for a gesture video stimulus was assigned based on the specific meaning used for that instance of communication, rather than in general for that gesture type. Six of the chimpanzee and 7 of the bonobo gesture types had a single meaning, and 4 chimpanzee and 3 bonobo gestures types had multiple meanings. For these ambiguous gesture types, participants viewed one instance where the correct outcome was the primary meaning (the most common recipient response to that gesture type), and one instance where the correct outcome was the alternate meaning (the second most common recipient response), and in both cases were given the primary and alternate meanings as possible answers. Some of the gesture types, for example, Directed Push, have different primary and alternate meanings, for example, "Climb on my back" for bonobos and "Move to a new position" for chimpanzees. For these, as for ambiguous gestures, we expect participants to answer with the correct response for the specific video.

## Results

A total of 17,751 people participated. We analysed $n = 112,648$ responses (Video only, $n = 59,001$; Context, $n = 53,647$) from $n = 5,656$ participants who completed the full set of videos (Video only, $n = 2,962$; Context, $n = 2,694$). Participants correctly interpreted the meanings of chimpanzee and bonobo gestures with or without additional Contextual information (Context: Success rate mean = $57.3 \pm 11.9\%$; binomial, $n = 53,647$, $p < 0.0001$; Video only: Success rate mean = $52.1 \pm 11.0\%$; binomial, $n = 59,001$, $p < 0.0001$) significantly higher than expected

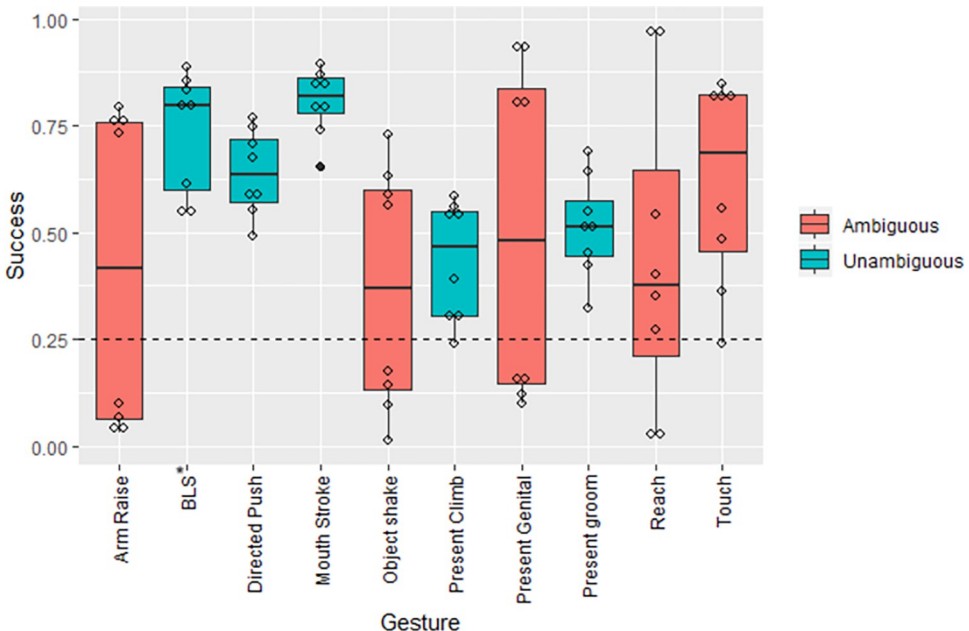

**Fig 1.** The distribution of correct responses for each gesture type, with the legend differentiating ambiguous and unambiguous gesture types (see https://greatapedictionary.ac.uk/gesture-videos2/ for illustrations and video examples of each gesture type; the data underlying this figure can be found at https://doi.org/10.5281/zenodo.7347608). *BLS, Big Loud Scratch.

by chance (0.25). Participants were above chance across all but one ("Object shake") gesture type (S1 Table and Fig 1).

Across gesture types, the addition of information on behavioural Context had an, at best, marginal positive effect on participant success (full-null model comparison: $X^2 = 5.746$, df = 2, $p = 0.057$; Table 1). More specifically, only the interaction between Context and Ambiguity showed any possible effect, with an again weak nonsignificant trend towards improved

**Table 1. Results of the glmm (estimates, standard errors, and significance tests).**

| Term | Estimate | Std. Error | Chisq | df | P |
|---|---|---|---|---|---|
| (Intercept) | 0.724 | 0.794 | | | [1] |
| Context_Video only | −0.054 | 0.222 | | | |
| Ambiguity_yes | −0.757 | 0.706 | | | |
| Context_Video only: Ambiguity_yes | −0.569 | 0.314 | 2.791 | 1 | 0.095 |
| Meaning_Climb on my back[2] | −0.571 | 0.741 | 5.649 | 7 | 0.581 |
| Meaning_Give me that food | 2.927 | 1.738 | | | |
| Meaning_Groom me | −0.302 | 0.751 | | | |
| Meaning_Let's be friendly | −0.680 | 0.891 | | | |
| Meaning_Let's have sex | 2.168 | 0.922 | | | |
| Meaning_Move away from me | 2.769 | 1.198 | | | |
| Meaning_Move into a new position | −0.606 | 1.008 | | | |
| Species_chimpanzee | 0.192 | 0.312 | 0.359 | 1 | 0.549 |
| z.Trial Number[3] | 0.031 | 0.037 | 0.684 | 1 | 0.408 |

[1]Not indicated because of very limited interpretability.

[2]Test statistic refers to the overall effect of meaning.

[3]z-transformed; mean and standard deviation of the original trial number were 10.490 and 5.765, respectively.

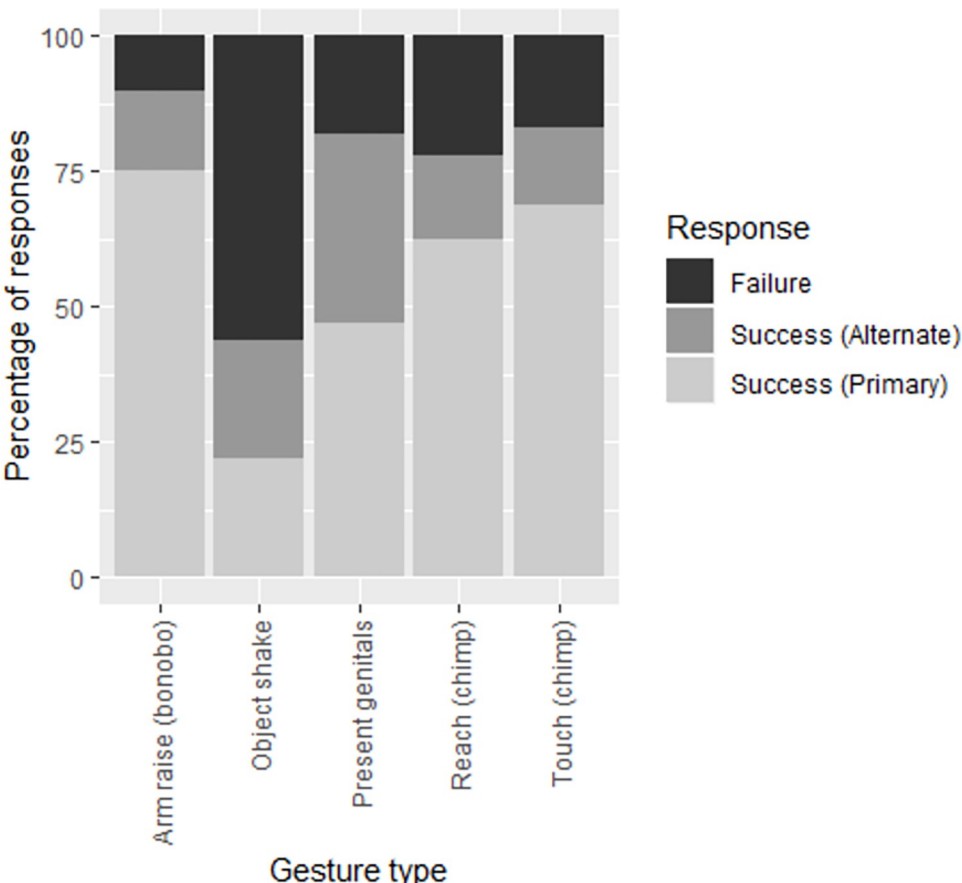

**Fig 2.** These five gesture types were ambiguous, having a primary (correct in this instance of use) and an alternate (correct in other uses) meaning included in the response options. This figure shows the percentage of responses for success selecting the primary meaning, success selecting the alternate, and failure to select either meaning for a given gesture type. The data underlying this figure can be found at https://doi.org/10.5281/zenodo.7347608.

participant success ($X^2$ = 2.791, df = 1, $p$ = 0.095) where gesture meaning was classed as Ambiguous and information on Context was available (S1 Fig and Table 1). Participants showed a small but significant increase in confidence in their responses for gestures with a single correct meaning (mean = 6.05, SD = 2.16), than for Ambiguous gestures (mean = 5.88, SD = 2.31; $t$ test: t(81,512) = −166, $p$ = 0.049).

Within Ambiguous gestures, where participants failed to select the correct meaning for this specific instance of communication (primary meaning), results were mixed as to whether they were more likely to select the secondary meaning for this gesture (alternate meaning, correct in other instances of use) than an incorrect meaning not associated with this gesture type (Fig 2). In 3 out of 5 gesture types, participants did not select the alternate meaning significantly above chance (S1 Table). Notably, the "Object shake" gesture is the only gesture type for which participants failed to assign either the primary or the alternate meanings.

## Discussion

Until now, humans have presented a problematic gap in the study of great ape gesture, with comparative observational methods limited to early development because of the feasibility of observing gesture production in humans after the onset of language [35]. By deploying a playback method that flips the paradigm from the study of gesture production to gesture

comprehension, we have accessed great ape gestural communication in adult humans for the first time. Participants were substantially above chance at assigning the "correct" meanings to chimpanzee and bonobo gestures across types, suggesting that humans may have retained their understanding of core features of a gestural system present in our last common ancestor with the *Pan* genus 6 to 7 million years ago [43]. This ability was present across both the functionally more fixed and the flexible gestures that are deployed with more than one meaning. Participants were highly successful at detecting the meaning for which gestures were used in the specific instance of communication that they saw. Where gestures had alternate meanings, these were also detected more often than chance in two gesture types. That our participants were able to interpret primate signals complements recent findings that suggest humans may be able to perceive affective cues in primate vocalisations [44].

The underlying mechanism that makes gestural communication comprehensible across great ape species, now including humans, remains unresolved. Humans use of gesture as intertwined with language in diverse ways makes detecting gesture types from the ape repertoire difficult. It remains unknown whether the great ape repertoire itself is biologically inherited [28], or whether apes—now including humans—share an underlying ability to produce and interpret naturally meaningful signals that are mutually understandable because of general intelligence and shared body plans and social goals, or the resemblance of gestures to the actions that they aim to elicit. These are not the only possible explanations, for example, gestures could be biologically inherited in nonhuman apes but understood by humans through other cognitive mechanisms, and we need to continue to develop innovative methods such as these video playbacks to address remaining unknowns.

Despite the importance of context in the interpretation of human communication [45] (and see [6] for bonobos), comprehension of great ape gestures was only marginally impacted by whether gestures had multiple meanings or whether participants were given the behavioural context in which the communication occurred. However, there were some gesture types for which we could not completely remove contextual information because it overlapped with gesture production, for example, the presence of food in some Mouth Stroke gestures. Future experiments with artificial stimuli may be able to test the limit of gesture comprehension by manipulating the amount and nature of information available, for example, stripping back situational context or exploring whether similar movements share a semantic core.

Our findings add a substantial new thread of evidence to the continuity of communication throughout our hominid lineage, and we propose that this novel citizen-science play-back approach will become a powerful and fruitful tool for bridging gaps in the study of comparative communication.

## Materials and methods

The experiment was presented in Gorilla.sc (www.gorilla.sc); a full preview and all importable sheets are available through Gorilla Open Materials (https://app.gorilla.sc/openmaterials/344409); and video data files are available at the Great Ape Dictionary on YouTube (https://tinyurl.com/greatapedictionary). Participants were recruited using a combination of online social and traditional media. The experiment ran from 20 July 2017 to 23 October 2017. The study was given ethical approval by the University of St Andrews University Teaching and Research Ethics Committee, under code PS12558.

### Participants

Each participant was asked for their year of birth to determine which version of the experiment they would be taken to, we had (a) a child-friendly game version for under-12s in which no

data were collected; (b) a child-friendly experiment for 12- to 15-year-olds; and (c) an adult experiment for participants aged 16 years and over. Our two experimental cohorts were then taken to age-appropriate consent forms. Where consent was provided, we collected demographic data about age (12 to 15, 16 to 20, 21 to 30, 31 to 40, 41 to 50, 51 to 60, 61 to 70, 71+), gender (Female, Male, Other, Prefer not to say), and experience working with nonhuman primates (No, Yes (0 to 2 years), Yes (3 to 5 years), Yes (over 5 years)), as well as whether the participants had done the experiment before (Yes, No). Demographic data were used to exclude participants who were too young to consent, who had experience with nonhuman primates, and who had done the experiment before. Participants were shown a set of instructions (Gorilla Open Materials: https://app.gorilla.sc/openmaterials/344409) before beginning the experiment. The adult group's experiment contained videos of gestures with the meaning "Let's have sex," while the child-friendly game version, and the adolescent group's experiment did not.

## Participant exclusions

A total of 17,538 over 15-year-olds and 213 twelve- to 15-year-olds participated. Data from the adolescent group (213), from adults who stated that they had any experience of working with nonhuman primates (480), from participants who stated that they had done the experiment before (143), and from participants who didn't complete the full experiment were excluded from the analyses (11,259).

## Design

Participants were randomly allocated to two conditions—those who viewed gesture videos only (Video only), and those who viewed gesture videos with a brief, one-line description of context (Gorilla Open Materials: https://app.gorilla.sc/openmaterials/344409). Each condition was divided into a further two groups, with each group being shown a different set of videos. We showed one example of each gesture type for both species (20 videos) to half of the participants, and a different example of each gesture type for both species to the other half (20 videos). Subgroups were split a final time into a further 4 random groups so that the position of the correct answer in each of the 4 box locations underneath the video varied among participants (Gorilla Open Materials: https://app.gorilla.sc/openmaterials/344409).

Videos were cut to show only the gesture, eliminating any behaviour before or after the signal. Each video showed the gesture once at regular speed and once in slow motion. Video lengths ranged from 7 to 33 seconds and could be watched as often as required before the answer was selected. A 500-millisecond fixation point was presented in the centre of the screen prior to each gesture video, videos were presented together with a Bonobo-bot illustration to highlight the gestural action within each video (Gorilla Open Materials: https://app.gorilla.sc/openmaterials/344409), and four possible meaning answers. The one-line descriptive text in the Context condition was presented below the video. After selecting an answer, participants were taken to a page and asked to rate their confidence in their answer using a sliding scale from not at all confident to 100% confident. At the end of the experiment, participants were provided with a numeric score, but no feedback on which questions were correctly answered.

We selected 10 gesture types for which we were previously able to confirm meaning in both chimpanzee and bonobos [19]. Gesture meanings were originally established using the Apparently Satisfactory Outcome: the response by the recipient that stopped the signaller from continuing to gesture [2,19]. Some gestures are used towards a single meaning, whereas others can be used towards two or more meanings. The correct meaning for a gesture video stimulus was

assigned based on the specific meaning used for that instance of communication, rather than in general for that gesture type.

For gesture types with a single primary meaning (gestures used towards a single meaning in 80% or more of cases; see S1 Table), both clips within a species showed gesture instances that went on to achieve that meaning (although note that this outcome could not be seen on the video stimuli presented). For gesture types with multiple meanings (each meaning used in at least 30% to 80% of cases; see S1 Table), one clip showed one meaning and one clip showed the other. For gestures regularly used with two meanings, the second meaning (incorrect for this specific instance of communication) was always included among possible answers.

The remaining response options were randomly selected from among the 8 meanings that were correct at some point in the experiment, and 3 meanings that are regularly achieved by apes with their gestures but not with the gesture types used in this experiment ("Follow me"; "Move closer to me"; "Stop doing that"). The answers were randomly selected, but if there was a repeat, we replaced it by skipping to the next randomly selected meaning so that an answer could only appear once among the 4 response buttons.

## Data exclusions

Data where all participant values were identical across all variables (Video only $n = 453$; Context $n = 368$) were eliminated as apparent upload error duplicates from page refreshing. Data where response time was shorter than 3 seconds (the minimum time required to watch the shortest real-time section of a gesture video) were excluded. Data where response time exceeded 3 standard deviations from the mean within each data set were also excluded.

## Data analyses

All analyses were conducted in R (version 3.5.3) [46]. We estimated the effect of gesture Context on participant success, by fitting a Generalised Linear Mixed model using the glmer function in package lme4 (version 1.1–27.1) with a binomial error structure and logit link function. We included Condition (Video only, Context), Ambiguity (yes, no), and their interaction, as well as Meaning, Species, and Trial number as fixed effects, and Participant ID, Gesture type, and Video ID as random effects. We included all possible random slopes, but correlations among random intercepts and slopes were not computationally feasible. As an overall test of the fixed effects, we compared the full model with a null model that was identical except for the exclusion of Condition. All significance tests were conducted using a likelihood ratio test [47].

The sample for this model included $n = 5,656$ Participant IDs, 40 Video IDs, and 10 Gesture types, with a total $n = 112,648$ responses. Prior to fitting the model, we z-transformed Trial number to a mean of 0 and a standard deviation of 1. All factors entering the random effects as random slopes were manually dummy coded and then centred. Confidence intervals were not computationally feasible.

## Full model results

We found a weak nonsignificant effect of Condition on participant success ($X^2 = 5.746$, df = 2, $p = 0.057$). More specifically, only the interaction between Condition and Ambiguity showed any possible effect, with an again weak nonsignificant trend towards improved participant success ($X^2 = 2.791$, df = 1, $p = 0.095$) where gesture meaning was classed as Ambiguous and information on Context was available (S1 Table).

## Supporting information

**S1 Table. Gesture types used, with the meanings for which they are used in both bonobos and chimpanzees.**
(DOCX)

**S1 Fig. Probability of success, separately for each combination of Context and Ambiguity.**
(DOCX)

## Acknowledgments

We thank the staff of the Budongo Conservation Field Station and the Wamba, Luo Scientific Reserve, where the gestural video data were collected, as well as the Ugandan National Council for Science and Technology, the Uganda Wildlife Authority, the WCBR, CREF, and the Ministère de la Recherche Scientifique et Technologie (Democratic Republic of Congo) for permission to work at these sites and in these countries. Video data were collected under UNCST Research Permit NS179 (Uganda) and Permis de Recherche No MIN.ESURS/SG-RST/002.2014 and Permis de Recherche No 002/MIN.RST/SG/180/002/2015 (DRC). We thank Professor Richard Byrne and Professor Takeshi Furuichi for their support and discussions. We thank Roger Mundry for support in running the code for analyses that would have otherwise been computationally nonfeasible.

## Author Contributions

**Conceptualization:** Kirsty E. Graham, Catherine Hobaiter.

**Data curation:** Kirsty E. Graham.

**Formal analysis:** Kirsty E. Graham, Catherine Hobaiter.

**Funding acquisition:** Kirsty E. Graham, Catherine Hobaiter.

**Investigation:** Kirsty E. Graham, Catherine Hobaiter.

**Methodology:** Kirsty E. Graham, Catherine Hobaiter.

**Project administration:** Kirsty E. Graham, Catherine Hobaiter.

**Resources:** Kirsty E. Graham.

**Software:** Kirsty E. Graham.

**Supervision:** Catherine Hobaiter.

**Validation:** Kirsty E. Graham, Catherine Hobaiter.

**Visualization:** Kirsty E. Graham.

**Writing – original draft:** Kirsty E. Graham, Catherine Hobaiter.

**Writing – review & editing:** Kirsty E. Graham, Catherine Hobaiter.

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
