## [Editor Report · Decision Letter 0]

30 Jun 2022

Dear Kirsty, 

Thank you for submitting your manuscript entitled "A great ape dictionary: language-using naïve humans understand non-human ape gestures" for consideration as a Research Article by PLOS Biology.

Your manuscript has now been evaluated by the PLOS Biology editorial staff, as well as by an academic editor with relevant expertise, and I'm writing to let you know that we would like to send your submission out for external peer review. Sorry for the delay incurred while we consulted an external expert.

IMPORTANT:

a) The Academic Editor found your study interesting, but was concerned about the interpretation and framing, and strongly recommends that you address these before we send it out for review. Specifically, the Academic Editor said:

"The authors conclude that humans understand the meaning of ape gestures and that this has implications for theories of language evolution. How exactly it relates to language is not entirely clear, and I would have preferred an actual study of how humans use hand gestures and interpret them in real life, but it is interesting that they understand ape gestures. They probably do so, not because they can reach back in evolutionary time, but because these gestures are still employed by our species.

"There exists a rich literature on ape gestures and multi-modal communication such as by Slocombe, Pollick, Liebal, Pika, Call, and others, but curiously little of it is mentioned here. Even the 2008 paper by Michael Corballis “The gestural origins of language” remains unmentioned.

"If the paper is shortened and the link with language de-emphasized (because unsubstantiated), and the paper focuses more on the universal nature of primate gestures to the point that humans understand those of apes, we may be getting close to a publishable paper. The study supports the shared gestural heritage of hominids in the same way that the literature often emphasizes the shared facial communication of hominids." [comments lightly edited]

b) I'm not convinced that you need to actually *shorten* your already concise paper, but you should address the above concerns and re-upload the manuscript when you upload the addition metadata (see next paragraph-but-one). Please also select "Short Reports" as the article type when you do so.

c) I've set your deadline to do all this as next Wednesday. If you need more time than this, do get in touch and we can discuss how to do that procedurally.

Once your full submission is complete, your paper will undergo a series of checks in preparation for peer review. After your manuscript has passed the checks it will be sent out for review. To provide the metadata for your submission, please Login to Editorial Manager (https://www.editorialmanager.com/pbiology) by Jul 06 2022 11:59PM.

Kind regards,

Roli

Roland Roberts, PhD

Senior Editor

PLOS Biology

rroberts@plos.org

---

## [Decision Letter · Decision Letter 1]

2 Sep 2022

Dear Dr Graham,

Thank you for your patience while your manuscript "A great ape dictionary: language-using naïve humans understand non-human ape gestures" went through peer-review at PLOS Biology. Your manuscript has now been evaluated by the PLOS Biology editors, an Academic Editor with relevant expertise, and by three independent reviewers (note that reviewer #1's comments are in an attachment).

You'll see that the reviewers are broadly positive about your study, but each of them has a number of requests to improve the manuscript. In particular, reviewers #1 and #2 have significant concerns about your interpretation of the results, and these are shared by the Academic Editor, who said [lightly edited]:

"I must agree with both reviewers 1 and 2 that something in the interpretation of results seems off. The study itself is impressive and well-conducted, and the reviewers have good suggestions for improvement that the authors should follow. But to argue that humans “retain” an understanding of gestural communication, as a reference to a hypothetical past in which our ancestors still used ape-like gestures, and the parallel claim that our current gestural repertoires barely overlap with those of the apes (the overlap is called “trivial”) seems problematic. As reviewer #2 writes, there is hardly any support for the last statement, and as reviewer #1 writes there are other ways of guessing what a gesture may mean. My own hunch is that human and ape gestural repertoires have plenty of overlap and that this explains the success of human subjects guessing gesture meaning, but there is also quite a bit of contextual help in this study to make these guesses. The overall interpretation of results needs to be revised to be more cautious about both a) what we know about human gestures compared to those of the apes and b) avoid references to a “retained understanding” and the implication that we humans have some evolutionary memory of past communication systems."

In light of the reviews, which you will find at the end of this email, we are pleased to offer you the opportunity to address the comments from the reviewers in a revision that we anticipate should not take you very long. We will then assess your revised manuscript and your response to the reviewers' comments with our Academic Editor aiming to avoid further rounds of peer-review, although might need to consult with the reviewers, depending on the nature of the revisions.

**IMPORTANT - SUBMITTING YOUR REVISION**

*Resubmission Checklist*

*Published Peer Review*

*PLOS Data Policy*

*Blot and Gel Data Policy*

Sincerely,

Roli Roberts

Roland Roberts, PhD

Senior Editor

PLOS Biology

rroberts@plos.org

REVIEWERS' COMMENTS:

Reviewer #1:

IMPORTANT: See attached file for review, which contains formatting and images!!!

Reviewer #2:

This was a very nice and interesting study reporting a novel citizen science experiment that examined people's comprehension of great ape (chimpanzee and bonobo) gestures. It was well written and well structured. The study used a simple but elegant design of presenting members of the public with videos of great ape gestures whose meanings were known, which they had to judge the meaning of in a forced-choice design. Participants were better than chance at judging the correct meaning of ape gestures, and interestingly context made only a relatively small difference. The authors are to be congratulated on the very impressive sample size: managing to engage over 17 000 particpants, of which 5600 provided useable data is a great achievement. Although assessing human perception in detecting animal signals is not itself new, this citizen science study is the first of its kind, including to explore it on such a large scale. The public engagement for the study is a great achievement. 

The study was well designed, well executed. Nevertheless, there are some aspects that warrant attention, particularly the framing, rationale as well as the interpretation. There are also a few omissions in the analyses which need attention (detailed below).

A key premise of the paper is that it is a conundrum that human gestures do not map onto great ape gestures (except in infants) (L65). There are two issues with this stance, First, the statement itself seems evolutionarily counter-intuitive, given evidence of extensive overlaps among other great ape gesture repertoires (making it most likely that humans overlap too), and there is also evidence of considerable continuities in the production and perception of communicative expressions of humans and other great apes in other modalities too (facial expressions and vocalisations). In this sense, it should be expected that humans can perceive gestures of apes too, above chance. This doesnt undermine their result, it just means its consistent with what one would expect. however, the statement that he producton of human gestures don't map onto ape gestures (at least from what is reported here) appears to be based on an absence of evidence, rather than evidence of absence. There is no study reported in the paper which addresses this, its only implied in various statements. Clarifying this is important as otherwise 'the conundrum' could more be just a case that production hasn't yet been adequately investigated. A study focussing on production would be a very exciting complement to this study. 

Also regarding framing, the paper starts out by resting on rather a strawman argument regarding a discontinuity between language and non-human communication (L30). As the authors themselves will be aware, evidence is revealing greater continuities than previously assumed. The authors contrast ape gestures to the apparently 'fixed' nature of primate alarm calls (L34-36); however there are multiple studies that now challenge this, for instance, studies of chimpanzee and orangutan alarm calls showing that caller's flexibly adjust their signals according to audience presence and knowledge (also potentially in other primates too e.g. audience effects in Thomas langur calls). Likewise, the next paragraph (L42) states that, by contast to animal signals, language 'does more than broadcast information'. Yet we now know that many animal signals also do more than broadcast information, as well as non-linguistic signals in humans. This part would warrant adjustment to avoid undermining progress that has recently been made to move away from such argumentation. 

The analogy made in L37-41 seems a little over-simplistic. Even in the absence of an audience, a signal may still be produced with some intention or be based upon a context which ordnarily would have a communicative goal. Linguistic signals also can also be used in this way, not just nonlingustic ones. E.g. When picking up a hot pan alone, one might still say "that was really hot!, That hurt.' - this may contain intentionality even in the absence of an audience. Moreover, there is also evidence that signalling to oneself may also serve other cognitive functions for the signaller. A signal even in the absence of an immediate audience may still have evolved to be communicative, even in the absence of an audience at that given moment. There is a lot to unpack here 

Methodology: The experimenters collected data on age, gender and experience with primates yet these variables didn't seem to be included in the analyses? It would be interesting and worthwhile to examine age and experience effects in particular, especially as the authors allude that perhaps human gestures get less 'ape-like' with age? It would also be interesting to see the demographic information about the sample population, was this collected?

Minor comments:

L33: One of the references used to support a sentence on primate alarm calls is actually about meerkats (Manser (2001)

L156: There is also evidence that humans can perceive the emotional content of primate calls, so this finding is not just limited to gestures. 

L236: For gestures with multiple meanings, there is a bit of a bias introduced in the experimental design which could increase chances to get at least one right if the 'second meaning' was always added as one of the options. To ensure chances remain at 0.25 per trial, it would have been good to have done a counterbalanced design where only one meaning option (first or second) was available. It's possible this was somehow controlled for and I missed it. 

Figures: The figures are nicely presented, just the fonts could be adjusted to be more legible

Table 1: Wasn't clear what the third column in the table was (unlabelled)

Reviewer #3:

[identifies herself as Heidi Lyn]

This article is an extremely interesting and modernized take on an older method of studying linguistic competence, comprehension. The authors present ape gestural data online and human participants indicate their understanding of the meaning of the gestures, to great success. I think the study is acceptable, with just a few clarifications to improve readibility, mostly.

Lines 92 - 96 - some examples of gestures with single and multiple meanings, as well as examples of primary and secondary meanings would be helpful. Also a description of how those meanings were determined.

Line 102 - why is the completion rate so low? Is it possible there is some bias introduced here?

Line 107 - are these binomials based on a 50% chance rate? But the participants chose from 4 possible answers? It would be helpful to clarify that chance would actually be much lower than 50% in the results, otherwise the percentage correct does not look impressive - because your numbers of trials are so high, a marginal improvement over chance would still be significant. Have you considered analyzing individuals' mean scores with a one-way t-test? Then you could also report on whether individual participants were able to recognize the entire repertoire.

Page 6 - I find both Figure 2 and Table 2 hard to interpret, although Figure 2 has me more puzzled. Is each gesture being added to the glmm as a separate predictor(Fixed effect)? Or a moderating variable? And you state there are 10 gestures, but only 7 meanings in the table? And their estimates in the glmm seem to vary widely from negative to positive. I'm not sure these visuals aid in the understanding of your findings or distract from them as the reader tries to interpret. 

Line 133 - this is a really interesting finding that I think should be emphasized more. When there was an alternate meaning, participants averaged closer to 75% choosing one of the possible meanings. In other words, choosing a completely unrelated meaning was much less likely than the initial percentages correct may suggest.

Discussion and Introduction: It would be great if there were some mention of the long history of comprehension as an earlier/better measure for language or symbolic competence in many species: e.g. 

Children: Shipley, E. F., Smith, C. S., & Gleitman, L. R. (1969). A Study in the Acquisition of Language: Free Responses to Commands. Language, 45(2), 322-342. https://doi.org/10.2307/411663), 

marine mammals: Herman, L. M., Richards, D. G., & Wolz, J. P. (1984). Comprehension of sentences by bottlenosed dolphins. Cognition, 16(2), 129-219. https://doi.org/10.1016/0010-0277(84)90003-9

and apes: Sevcik, R. A., & Savage-Rumbaugh, E. S. (1994). Language comprehension and use by great apes. Language & Communication, 14(1), 37-58.

An Extended Data Table is mentioned several times, but I don't see it?

---

## [Decision Letter · Decision Letter 2]

17 Nov 2022

Dear Dr Graham,

Thank you for your patience while we considered your revised manuscript "Towards a great ape dictionary: naïve humans understand non-human ape gestures" for publication as a Short Report at PLOS Biology. This revised version of your manuscript has been evaluated by the PLOS Biology editors, the Academic Editor, and two of the original reviewers.

Based on the reviews and our Academic Editor's assessment of your responses to reviewer #2 (who was not able to re-review), we are likely to accept this manuscript for publication, provided you satisfactorily address the remaining points raised by the reviewers. Please also make sure to address the following data and other policy-related requests.

IMPORTANT:

a) Regarding the Title, 1. we tend to avoid punctuation in Titles, so we suggest that you remove the first five words (though we would understand if you wanted to keep them); 2. You'll see that reviewer #1 thinks that including "some" in the Title would be more accurate, and you should consider this; 3. One of my colleagues felt that "naïve" often has pejorative connotations, and should be avoided; however, I couldn't think of a suitable substitute.

b) Please attend to the remaining points from reviewer #1.

c) Please address my Data Policy requests below; specifically, we need you to supply the numerical values underlying Figs 1, 2 and S2, either as a supplementary data file or as a permanent DOI’d deposition. I also note that your main data and code deposition is in Github; please could you make a permanent DOI’d copy (e.g. in Zenodo) and cite this URL in the paper?

d) Please cite the location of the data clearly in all relevant main and supplementary Figure legends, e.g. “The data underlying this Figure can be found in S1 Data” or “The data underlying this Figure can be found in https://doi.org/XXXX”

We expect to receive your revised manuscript within two weeks. 

*Published Peer Review History*

*Press*

Sincerely,

Roli Roberts

Roland Roberts, PhD

Senior Editor,

rroberts@plos.org,

PLOS Biology

DATA POLICY:

Regardless of the method selected, please ensure that you provide the individual numerical values that underlie the summary data displayed in the following figure panels as they are essential for readers to assess your analysis and to reproduce it: Figs 1, 2 and S2. NOTE: the numerical data provided should include all replicates AND the way in which the plotted mean and errors were derived (it should not present only the mean/average values).

DATA NOT SHOWN?

REVIEWERS' COMMENTS:

Reviewer #1:

I am happy to recommend publication of this very, very interesting paper (kudos to the authors for paying great attention to the objections), but I would suggest that some additional clarifications/improvements be made.

1. Optionally: ***this should be left to the authors***, but personally I would prefer a title that's less generic, e.g. with 'some': e.g. "Towards a great ape dictionary: naïve humans understand some non-human ape gestures."

The generic suggests that naïve humans *generally* understand ape gestures, and this seems to me to be an overstatement. Something less generic would be more accurate given the small number of gestures investigated. I do understand that this might take away from the impact of the article, so I'll understand if the authors prefer something that makes a bigger splash (although in the long term, I am not sure this is the best strategy).

2. I would recommend that the abstract be rewritten to state clearly and explicitly that humans understand (some) ape gestures, but that this could be for 2 reasons: 1. an innate communication system has been retained. 2. general intelligence allows humans to guess the meaning of some ape gestures.

Again, this might weaken the short-term impact of the paper, but it's a far more accurate assessment of the finding and will help with the long-term credibility of this (wonderful) line of research. 

3. Still in the abstract, I think the authors could say in so many words that the 10 gestures studied were the most common ones. This strengthens the authors' point, as this is a clear criterion, which might defuse the reader's worry that the choice of items was biased.

4. The final discussion could be clarified. The following is a great improvement:

"It remains unknown whether the great ape repertoire itself is biologically inherited (Byrne et al. 2017), or whether apes - now including humans - share an underlying ability to produce and interpret naturally meaningful signals that are mutually understandable because of general intelligence and shared body plans and social goals, or the resemblance of gestures to the actions that they aim to elicit"

But I think this is still insufficiently accurate. This reads as if there are 2 salient theoretical possibilities.

Possibility 1: In apes and humans alike, the ape repertoire is (genetically) inherited.

Possibility 2: In apes and humans alike, the ape repertoire is understandable because of general intelligence etc.

But this incorrectly suggests that the most salient theories are ones in which apes and humans share an ability in this connection. This is not correct, as a very salient possibility is a third one, namely:

Possibility 3: In apes, the ape repertoire is (genetically) inherited. In humans, the ape repertoire is understood through general intelligence etc.

Possibility 3 is a deflationary account of the authors' finding. It's useful to mention it very clearly because it might trigger further research to prove or disprove the point.

5. The term 'meaning'

I appreciate the authors' response to my earlier suggestion, copied at the bottom of this report. But I think it would be easier if they stated clearly and explicitly at the beginning of the paper something along the following lines:

In this article, we use the term 'meaning' to refer to a reaction type, specifically to what has been called an 'apparently satisfactory outcome' (ASO) in the earlier literature.

with appropriate references to the literature, and examples. As I wrote earlier, without such a definition, linguists will be baffled by this use of the term 'meaning'. And it just takes a few clear words to provide a clear definition.

6. Human playback experiment

The description copied below (a version of which appears in the discussion) has the advantage of coolness ('performing ape-like playback experiments with humans, what a neat idea!'). But I think it doesn't do justice to the experimental situation. First, the authors employ a standard methodology of psychology/psycholinguistics: the subjects are just asked to guess the meaning of various things in perception. Second, no playback experiment with animals would look anything like this, as the human subjects are asked to do something entirely unnatural (they need to read complex instructions and provide guesses about meanings/functions). I would recommend restating all this far more simply. The coolness factor might decrease, but credibility would increase.

"We employ a method regularly used in studies of non-human primate communication, a 'play-back' experiment, in which recipient behaviour is analysed following exposure to a signal (Fischer et al. 2013, Radick 2005). This type of comprehension study has historically been employed to test nonhuman species on comprehension of human language (Herman et al. 1984, Sevcik & Savage-Rumbaugh 1994), but here we flip the paradigm to test humans on nonhuman communication. While language-competent humans seem to no longer typically produce"

As before, I think this is a great experiment and a super interesting paper!

EARLIER SUGGESTION ABOUT 'meaning'

Meaning: The term 'meaning' is used in the paper for 'Apparently Satisfactory Outcomes' (ASOs). This should be clarified in the paper. To anyone familiar with linguistics, it seems entirely implausible that ASOs are meanings, i.e. the cognitive representations of the informational content of gestures. Rather, ASOs are convenient ways to operationalize the various uses of gestures, but it seems clear that the core meanings of these gestures haven't been discovered yet. This is not a problem for the present study, but some readers will be misled if the authors do not state clearly what they understand by 'meaning' (namely ASOs).

Thank you, that's an important distinction that we have now clarified on Line 106 "We selected the 10 most common gesture types for which we were previously able to confirm "meaning" in both chimpanzees and bonobos, determined by recipient responses that consistently satisfy the signaller (Graham et al. 2018)"

Reviewer #3:

The authors have done an excellent job in responding to the reviewers' concerns. I recommend acceptance.

---

## [Editor Report · Decision Letter 3]

30 Nov 2022

Dear Dr Graham,

Thank you for the submission of your revised Short Report "Towards a great ape dictionary: inexperienced humans understand common non-human ape gestures" for publication in PLOS Biology. On behalf of my colleagues and the Academic Editor, Frans de Waal, I'm pleased to say that we can in principle accept your manuscript for publication, provided you address any remaining formatting and reporting issues. These will be detailed in an email you should receive within 2-3 business days from our colleagues in the journal operations team; no action is required from you until then. Please note that we will not be able to formally accept your manuscript and schedule it for publication until you have completed any requested changes.

Sincerely,

Roli Roberts 

Senior Editor

PLOS Biology

rroberts@plos.org